# Image Analysis in Histopathology and Cytopathology: From Early Days to Current Perspectives

**DOI:** 10.3390/jimaging10100252

**Published:** 2024-10-14

**Authors:** Tibor Mezei, Melinda Kolcsár, András Joó, Simona Gurzu

**Affiliations:** 1Department of Pathology, George Emil Palade University of Medicine, Pharmacy, Science, and Technology of Targu Mures, 540139 Targu Mures, Romania; simona.gurzu@umfst.ro; 2Department of Pharmacology and Clinical Pharmacy, George Emil Palade University of Medicine, Pharmacy, Science, and Technology of Targu Mures, 540142 Targu Mures, Romania; melinda.kolcsar@umfst.ro; 3Accenture Romania, 540035 Targu Mures, Romania; andras@onezerozero.eu

**Keywords:** digital image analysis, microscopy, pathology, cytopathology

## Abstract

Both pathology and cytopathology still rely on recognizing microscopical morphologic features, and image analysis plays a crucial role, enabling the identification, categorization, and characterization of different tissue types, cell populations, and disease states within microscopic images. Historically, manual methods have been the primary approach, relying on expert knowledge and experience of pathologists to interpret microscopic tissue samples. Early image analysis methods were often constrained by computational power and the complexity of biological samples. The advent of computers and digital imaging technologies challenged the exclusivity of human eye vision and brain computational skills, transforming the diagnostic process in these fields. The increasing digitization of pathological images has led to the application of more objective and efficient computer-aided analysis techniques. Significant advancements were brought about by the integration of digital pathology, machine learning, and advanced imaging technologies. The continuous progress in machine learning and the increasing availability of digital pathology data offer exciting opportunities for the future. Furthermore, artificial intelligence has revolutionized this field, enabling predictive models that assist in diagnostic decision making. The future of pathology and cytopathology is predicted to be marked by advancements in computer-aided image analysis. The future of image analysis is promising, and the increasing availability of digital pathology data will invariably lead to enhanced diagnostic accuracy and improved prognostic predictions that shape personalized treatment strategies, ultimately leading to better patient outcomes.

## 1. Introduction

Pathology is one of the more traditional branches of modern medicine, encompassing histopathology, cytopathology, and molecular pathology. For many years, morphology stood at the heart of pathology, including not only features visible to the naked eye but also further details seen under the microscope [1]. Although gross appearance continues to be very important, the histological slide and cytological smear have evolved as the two main diagnostic mediums of the pathologist.

Histopathology and cytopathology have stood the test of time, and despite many advances in both fields, they are still largely based on recognizing microscopical morphologic patterns. Both are quite conservative in terms of preparation techniques, as the stains used to highlight various tisular and cellular features have remained quite unchanged over the past century or so. The hematoxilin and eosin staining technique was introduced in the late 19th century and is still used on a daily basis [2,3,4]. Cytopathology is no exception to the rule; its main staining methods, developed several decades ago, are still being used to this day [5,6]. Their universal adoption and global success are probably attributable to their unique properties to highlight morphological changes that aid the identification of various pathological entities [4]. For many years, the human brain was the exclusive ‘tool’ to ‘compute’ a diagnosis from visual information of what the eyes saw under the microscope. Our huge body of histopathological and cytopathological medical literature is largely based on this information that was passed on, and still is, to several generations of pathologists. Human diagnostic recognition comes from ever-growing personal experience that stems from seeing similar or identical lesions. There is an enormous amount of information present within microscopic images that the human eye can perceive and the human brain processes. Additionally, interpolation of microscopic features with the patient’s clinical history and lab data was always relevant [7,8].

The advent of computers and digital imaging technologies came to challenge the exclusivity of human eye vision and brain computational skills and carved their way into the diagnostic process, both in histopathology and cytopathology. From the early days of manual microscopic observations to the current era of sophisticated digital imaging and computational analysis, we have witnessed a remarkable progression in the tools and methods used to study and quantify cellular and tissue morphologies [9,10].

### 1.1. Image Processing and Analysis in Medical Diagnostics

There are several medical fields that process visual data, including, but not limited to, radiology and pathology. The purpose of this processing, most of the time, revolves around establishing a diagnosis or extracting other useful information that is relevant for patient management [11,12,13].

In both histopathology and cytopathology, the diagnostic process involves the interpretation of a large number of microscopic images [14]. For example, to fully examine a 1 cm^2^ tissue section under the microscope with the 10× objective (with a field of view of 18 mm), one needs to check nearly 400 fields, but that number is close to 50,000 if the objective is 40×, so that is quite a lot of visual data. Despite this, the trained human eye knows exactly what to look for, and the human brain can discriminate between what is important and needs to be checked at higher magnification and what is not, and, therefore, can be scanned at lower magnification. But the human brain is not perfect. A recent study found that cognitive bias can be a source of error in diagnostic pathology, especially in tumor grading [15].

Computers, on the other hand, can deal with large amounts of data, and current trends point to a future where they will be able to make a diagnosis, or—to comfort the skeptics—at least *aid* the pathologist in making a diagnosis.

### 1.2. Objectives and Scope of the Review

The goal of this review is to provide an overview of the development of digital image processing methods specifically utilized in pathology and cytopathology. Some clinical applications are also discussed, touching on subjects such as ethics and regulatory concerns. Finally, future perspectives are also discussed.

## 2. Historical Developments

The human mind has its own *modus operandi* for interpreting various microscopic features. This was partially reflected in the way some of the lesions were described and identified. Often, the nomenclature we use to describe certain microscopic details contains, even to this day, the names of everyday objects to highlight more sophisticated changes or features; expressions such as herringbone pattern, lepidic pattern, cribriform, oat-cell carcinoma, tadpole cells, flame cells, polka dot cells, coffee-bean nuclei, and Orphan Annie nuclei, and so on, are some salient examples, but the list is numerous. Image analysis came to give mathematical meaning to these descriptors.

Manual image analysis methods (e.g., visual inspection, planimetry, manual counting) has historically been the primary approach in histopathology and cytopathology, relying on pathologists’ expert knowledge and experience to visually examine and interpret microscopic tissue samples [7]. To aid them in the process, special microscopy tools, such as eyepiece graticules or microscope calibration slides, were used to attain precision when measuring distance or area in the microscopic field. However, these processes were time-consuming, subject to the subjective biases of the individual pathologist, and unable to efficiently process large volumes of data [16]. The increasing digitization of histopathological images has paved the way for the application of more objective and efficient computer-aided analysis techniques.

Early automated image analysis methods in pathology and cytopathology were developed to address the limitations of manual microscopic evaluation. These early automated techniques took advantage of digital imaging advancements and aimed to automate tasks such as cell counting, feature quantification, and pattern recognition. However, these initial automated methods were often constrained by the computational power available at the time as well as the complexity of biological samples, which posed significant challenges in accurately segmenting and analyzing cellular and tissue structures. Even though the word ‘automation’ was used, these techniques assumed that digital photomicrographs were taken by the pathologist and, later, by the microscopes mounted with motorized stages.

To achieve specific tasks, in the early days of digital pathology, a specific code was written tailored to identify various histological features. The use of specific staining methods (other than hematoxylin and eosin) facilitates thresholding [17].

The last decade has witnessed a surge in the application of more advanced machine learning (ML) and deep learning (DL) algorithms to the analysis of medical images, particularly in the fields of histopathology and cytopathology. In 1955, John McCarthy and his colleagues pioneered the concept of ‘artificial intelligence’, (AI) a term that has only recently permeated public consciousness with the advent of large language models and their availability to the large public [18]. The term ‘deep learning’ was used first by Rina Dechter in a 1986 paper [19]. To best illustrate the interest of the academic community in these topics, we performed several queries on the PubMed database using various search terms (Figure 1 and Figure 2). The number of published articles parallels the development of the digital image analysis arsenal, including the development of convolutional neural networks and deep learning.

These techniques have shown great promise in automating and enhancing various aspects of image analysis, including cell detection, segmentation, classification, and quantification. Machine learning algorithms have been successfully employed in the analysis of histopathological images, leading to advancements in areas such as disease diagnosis, prognosis, and patient stratification. These algorithms can be trained on large datasets of annotated histopathological images to learn complex patterns and features that may be subjected to interobserver variability among pathologists [20]. The advancement of hardware was paralleled by the development of more sophisticated and optimized algorithms, paving the way for complex image analysis of tissues and cells. A brief timeline is presented with selected publications that significantly impacted the advancement of digital image analysis (Figure 3), with more details in Section 3.2 [21,22,23,24,25,26,27].

The use of supervised learning techniques, such as convolutional neural networks (CNN), for tasks like tumor detection, grading, and proliferation index scoring in breast cancer were, for example, employed [28,29]. These models have demonstrated the ability to provide more consistent and accurate assessments compared to manual analysis by pathologists.

Cytopathology also benefited from the integration of computational techniques. Automated cell segmentation, feature extraction, and classification algorithms have been developed to assist in the analysis of cytological samples, such as those obtained from fine-needle aspirations or body fluids. These computational approaches have the potential to enhance the efficiency and reproducibility of cytopathological analysis, particularly in scenarios where the volume of samples can be overwhelming for manual review [30,31,32].

## 3. Current Techniques and Technologies in Image Analysis

Image analysis in pathology and cytopathology has undergone significant advancements, particularly with the integration of digital pathology, ML, and advanced imaging technologies that have revolutionized image analysis in these fields, enhancing diagnostic accuracy, efficiency, and research capabilities [33].

### 3.1. Digital Pathology

Digital pathology involves the acquisition, management, and interpretation of pathology information in a digital environment. This transformation from traditional microscopy to digital platforms has opened new possibilities in tissue analysis, data sharing, and remote diagnostics (telemedicine). Furthermore, published data suggest that digital slides are not inferior to conventional glass slides for primary diagnosis [34]. Additionally, they are suitable for cytopathology applications as well [35].

#### 3.1.1. Whole Slide Imaging

Whole slide imaging (WSI) technology has enabled the digitization of physical tissue slides by scanning, creating high-resolution digital images that can be viewed and analyzed on a computer [36]. The development of digital scanners enabled the use of multiple magnifications and focal planes (z axis), leading to the accumulation of large datasets of digital histopathological images, which can be leveraged to train machine learning models for automated analysis and interpretation. Thus, WSI has become the cornerstone of digital pathology. Furthermore, this technology allows for remote consultations and facilitates the use of image analysis algorithms to quantify and classify tissue features automatically. Recent advancements in WSI technology have improved image resolution and scanning speed, making it a practical tool in clinical settings. There is an excellent white paper on the matter written by Zarella et al. [37,38,39].

Kim et al. summarized the results of the American Society of Cytology (ASC) Digital Pathology White Paper Task Force survey and concluded that cytopathology is somewhat behind surgical pathology in terms of adopting WSI into clinical practice. This is partly due to the fact that cytological smear interpretation requires exceptional image quality to assess morphological features. This is partly the reason why cytological smears are preferentially scanned with a 40× objective, rather than a 20× objective [40].

#### 3.1.2. Image Storage and Retrieval Systems

An average digital slide, depending on the scanning properties and settings, may reach significant sizes that easily exceed 30 Gbytes per slide. Therefore, efficient storage and retrieval systems have become essential. These systems, often integrated with Laboratory Information Management Systems (LIMS), support the organization and retrieval of vast amounts of image data. Cloud-based storage solutions offer scalability and data accessibility, which are crucial for collaborative research and telepathology.

Automated image analysis involves the use of computational methods to interpret and quantify features in digital pathology images. This automation reduces subjectivity and improves the reproducibility of diagnostic assessments. The increasing availability of digital pathology has necessitated the development of robust image storage and retrieval systems. These systems allow for the efficient management and organization of large volumes of medical image data, facilitating access and analysis by clinicians and researchers.

### 3.2. Digital and Automated Image Analysis

The application of ML and DL algorithms to the analysis of digital pathology images has led to significant advancements in the field of automated image analysis. These techniques have demonstrated the ability to perform tasks such as cell segmentation, tissue classification, and prognostic prediction with high accuracy, potentially enhancing the efficiency and objectivity of pathological assessment [41].

#### 3.2.1. Segmentation Techniques

Segmentation is a critical step in manual or automated image analysis, where specific structures such as cells, nuclei, or tissues are delineated from the background. It involves the partitioning of digital images into multiple segments or regions. Automated image segmentation is a subfield of computer vision and digital image processing that seeks to categorize similar regions or segments of an image according to their respective class labels. Image segmentation takes image classification a step further by incorporating localization alongside classification. Image segmentation can be characterized by the *tasks* it aims to perform and by the *methods* it utilizes.

Image segmentation **tasks** vary according to the desired goal and are dependent on the amount and type of information that is to be extracted from images. Types of segmentation tasks include semantic segmentation, instance segmentation, panoptic segmentation, and medical image segmentation. Medical image segmentation tasks may be very diverse, making the generalization of existing methods difficult [42,43]. The SAM (Segment Anything Model) AI model created by Kirillov et al. showed promising results in digital pathology applications [44,45]. The MedSAM model developed by Ma et al. aims to overcome the lack of generalized applicability of existing models, offering universal medical image segmentation [43].

Image segmentation **methods** may be grouped as traditional and deep learning models. Both are using pixel color values and associated characteristics, such as brightness, contrast and intensity. More traditional methods include thresholding, edge detection, watershed, and region- and clustering based segmentation. Deep learning models use CNNs. Unet is a notable example, as it was specifically designed for medical applications [27,46,47,48,49]. The more traditional segmentation algorithms were described long before the robust computational power of today’s computers was available [21,22,23]. Some of the traditional segmentation techniques are shown in Figure 4 and Figure 5. As there is a growing number of segmentation techniques, especially using AI methodology, it is recommended to measure and report three metrics, accuracy, precision, and efficiency, to evaluate their performance [50].

Convolutional neural networks (CNNs, *see below*) have been particularly effective in accurately segmenting complex histological structures, which is essential for subsequent feature extraction and classification tasks [52]. The primary task of these techniques is the extraction of quantitative information about the size, shape, and spatial distribution of various elements within the sample.

One of the key challenges in digital pathology is the accurate segmentation of individual cells or tissue structures within complex and heterogeneous histopathological images that the human eye and brain can do so well on individual images. Segmentation algorithms based on deep learning specifically developed to address this challenge enable precise delineation of cellular and subcellular features for further analysis [36].

#### 3.2.2. Feature Extraction and Quantification

After segmentation, the next step is feature extraction, where measurable characteristics such as size, shape, texture, and intensity are quantified. This process includes the systematic extraction of meaningful information, such as the size, shape, texture, and spatial distribution of cellular and subcellular structures, from the digital image data. Once the relevant structures have been segmented (e.g., isolated or delineated), the next step is to extract and quantify their morphological and textural features. This information can be used to train machine learning models for tasks such as disease diagnosis, prognosis, and personalized treatment planning [53]. These features are also crucial for distinguishing between normal and pathological tissues [54]. For instance, nuclear pleomorphism, a key feature in cancer diagnosis, can be quantified automatically, providing objective metrics that complement traditional histopathological assessments [55].

Quantitative image analysis and precise morphometric measurements have become pivotal to the field of histology and cytology [56]. Advanced computational techniques are based on the development of landmark algorithms that enable the systematic quantification of visual features within images. These techniques not only improved the speed of image processing but also are able to extract features with less error than previous algorithms. Examples include Scale Invariant Feature Transform (SIFT), developed by Lowe [24], and Speeded-Up Robust Features (SURF), developed by Bay et al. [25]. See also Figure 3.

#### 3.2.3. Classification and Pattern Recognition

Classification and pattern recognition algorithms have become increasingly sophisticated and pivotal in the field of digital image analysis for histology and cytology [57].

These advanced computational techniques enable the automated identification, categorization, and characterization of different tissue types, cell populations, and disease states within microscopic images [58]. Through the application of ML and DL methods, classification models can be trained to recognize complex visual patterns and features that are indicative of specific pathological conditions or cellular phenotypes [59,60]. This allows for more accurate, objective, and scalable analysis of histological and cytological samples. One of the most cited papers came from Krizhevsky et al., describing an algorithm that performed very effectively in the ImageNet contest, achieving 15.3% error rate, significantly better than other entries in respect to image classification [26].

### 3.3. Artificial Intelligence and Machine Learning in Image Analysis

The application of ML and AI has brought about a paradigm shift in histopathology and cytopathology image analysis. These technologies enable the development of predictive models that can assist in diagnostic decision making [28]. These approaches have demonstrated the ability to outperform human experts in certain tasks, such as the identification of specific cell types or the prediction of disease outcomes. These advancements look back to a recent but formidable history, which included those challenges that required categorization of multiple images [61,62,63].

ML methods are broadly categorized as supervised and unsupervised learning, and, in medical imaging, rarely used, other types like semi-supervised learning and reinforcement learning [64].

Conventional AI techniques for histopathology image analysis have concentrated on refining specialized models for individual diagnostic tasks. Despite achieving some success, they lack generalizability. A weakly supervised ML framework was designed to overcome this, called Clinical Histopathology Imaging Evaluation Foundation (CHIEF) model, which promises the extraction of pathology imaging features for systematic cancer assessment [65].

AI-driven tools are being increasingly integrated into clinical workflows and have the potential to assist pathologists in their daily workflow, reducing the risk of human error and improving the consistency and accuracy of diagnoses [66]. Decision support may be achieved by highlighting areas of concern on digital slides, suggesting potential diagnoses, and even predicting patient outcomes based on histological features. The adoption of these tools is still in its early stages, but they hold promise for enhancing diagnostic accuracy and reducing workload in pathology departments [39].

Convolutional neural networks (CNNs) are a key component of AI architectures, particularly in the field of machine learning, or more specifically, deep learning. CNNs are inspired by the way the brain processes visual information. Nevertheless, the analogy is not perfect because it simplifies many aspects of biological vision, focusing on hierarchical and local feature extraction rather than the more efficient, adaptable, and dynamic processing that characterizes the brain. This may change with further developments in the field. CNNs have shown remarkable success in image classification tasks, including cancer detection and grading.

These models are trained on large datasets of annotated images, allowing them to learn complex patterns and features that may be difficult for the human eye to discern [67,68]. For example, deep learning models have been developed to classify breast cancer subtypes based on histological images, achieving accuracy levels comparable to those of experienced pathologists [69]. For instance, some studies suggest that deep learning algorithms are more accurate in detecting lymph node metastases [70,71]. Cervical cytopathology also may benefit from image analysis that uses deep learning techniques [72]. Such tools have been explored in various areas, including the assessment of mitotic index in breast cancer, the prediction of molecular profiles in lung and gastrointestinal cancers, and the diagnosis of prostate cancer [73].

The creation of training datasets assumes the creation of annotated digitized images, often hindered by the low availability of human annotators. This led some researchers to develop an approach that trains diagnostic CNN models without the use of human annotations [74].

A recent study found that AI algorithms are not widely used in everyday practice; nevertheless, there is a growing interest in the use of these technologies both in histopathology and cytopathology applications. Currently, academic centers are the flagships of implementation [40,75].

### 3.4. 3D Imaging and Advanced Visualization Techniques

Traditional histology relies on 2D tissue sections, but recent advancements in 3D imaging are providing more comprehensive views of tissue architecture [76]. The introduction of 3D imaging modalities, such as confocal microscopy and light sheet fluorescence microscopy, has enabled the acquisition of high-resolution, 3D representations of tissue samples. This has opened up new possibilities for the analysis of complex cellular and tissue structures, as well as the investigation of spatial relationships between different cell types and subcellular components [77].

Three-dimensional reconstruction involves the creation of 3D models from serial sections of tissue. The analysis of 3D images requires specialized techniques for image reconstruction, segmentation, and quantification. Several algorithms have been developed allowing for the extraction of precise morphometric data and the exploration of spatial relationships within tissue samples. Techniques such as *optical projection tomography (OPT)* and *light sheet fluorescence microscopy (LSFM)* enable the visualization of tissue structures in 3D, offering new insights into tissue organization and pathology. These methods are particularly useful in research settings for studying complex tissues such as the brain or tumors. Three-dimensional imaging has applications in both research and clinical practice. In cancer research, for example, 3D models of tumors provide a more accurate representation of tumor morphology, which can aid in understanding tumor invasion and metastasis.

### 3.5. Virtual Staining and Deep Learning

Deep learning enabled the development of virtual staining methods that replicate traditional staining from unlabeled histological samples [78,79,80]. These techniques have the potential to replace traditional chemical-based staining methods, therefore avoiding their downsides (toxicity, inconstant tinctorial characteristics, etc.) [81,82,83]. To achieve this, however, excellent quality and large datasets are required to train specific models.

## 4. Applications in Clinical Diagnostics

### 4.1. Histopathological Image Analysis for Cancer Detection and Grading

Automated image analysis algorithms can be used to identify and quantify specific cellular and morphological features that are indicative of malignancy, enabling a more accurate and consistent diagnosis [54]. These studies have demonstrated the use of deep learning models to predict the Ki-67 proliferation index in breast cancer, a key prognostic marker, directly from histopathological images. Applications include pathologies such as breast cancer [29,84], prostate cancer [59,73], lung cancer [34], gastric cancer [85], and urothelial carcinoma [86].

Analyzing histological images of tissue samples is a crucial step in the diagnosis and staging of various types of cancer. ML algorithms can be trained to identify and segment specific cellular and tissue structures, enabling the quantification of features such as nuclear size, shape, and density, which are often used to assess the grade and stage of a tumor [20,73,77]. The application of deep learning, in particular, has shown promise in automating tasks such as the identification of mitotic figures, the classification of different cancer subtypes, and the prediction of molecular profiles from histopathological images. Computational pathology is the term used to describe the use of AI tools to analyze tissue sections [87,88,89,90].

AI-driven image analysis was successfully used in one study to identify lymph node metastasis in breast cancer. A sensitivity of 100% and a specificity of 78.5% were obtained, with a positive predictive value of 68.1% and negative predictive value of 100%. This task has a shorter time to perform, making it a potential tool for screening purposes [91]. The deep learning-based algorithm (LYmph Node Assistant/LYNA) for detection of metastatic breast cancer in sentinel lymph nodes was not affected by histology artifacts related to overfixation, staining quality, and air bubbles [70,92]. One study found that AI-assisted pathologists were more accurate in diagnosis than either the algorithm or the pathologist alone; algorithm assistance improved the sensitivity of micrometastasis detection from 83 to 91% [93]. Large-scale image analysis was successfully applied for Gleason score assessment in prostate biopsies, resulting in an accuracy of 75% in the Gleason score assessment [94].

The quantification of established biomarkers, such as the PD-L1 assessment, would also greatly benefit from automated and objective scoring, given the therapeutic consequence of the assessment [95].

### 4.2. Cytological Image Analysis for Early Detection of Disease

In addition to histological analysis, the examination of cytological samples, such as those obtained through fine-needle aspiration or exfoliative cytology (e.g., cervix cytology or others), can also provide valuable information for the early detection of cancer. Automated image analysis techniques can be used to identify and characterize individual cells, detecting subtle changes in morphology that may be indicative of malignancy. These approaches can enhance the sensitivity and specificity of cytological screening, leading to earlier diagnosis and improved patient outcomes [96].

Computer-aided diagnosis (CAD) systems, including ImageJ software, have significantly enhanced the early diagnosis of thyroid cytopathology. These technologies facilitate improved image analysis and interpretation, leading to more accurate thyroid nodules and lesions assessments. ImageJ software, an open-source image processing tool, allows for the advanced quantitative analysis of cytological samples obtained through fine needle aspiration (FNA). Analysis of various cytomorphometric parameters can enhance diagnostic accuracy by providing objective measurements that reduce interobserver variability [97]. This is particularly important in the context of the Bethesda System for Reporting Thyroid Cytopathology (BSRTC), which categorizes FNA results into distinct diagnostic categories, thereby facilitating standardized reporting and management [98]. Integrating analysis of multiple imaging modalities, such as ultrasound and cytological samples, can further improve diagnostic confidence and accuracy, particularly for nodules with intermediate risk of malignancy. This enables clinicians to make more informed decisions regarding patient management and treatment options.

Furthermore, the automation of image analysis reduces the time required for diagnosis, allowing quicker clinical decision making. Digital morphometry could be used to develop a computer-aided tool for assisting the pathologist in diagnosing thyroid nodules [99,100,101].

Digital image analysis in cytopathology practice has been significantly pioneered by the PAPNET system, being the first such system approved by the FDA [102]. Others followed (BD FocalPoint Slide Profiler, ThinPrep Imaging System), and published data indicate that these systems perform well in terms of sensitivity and specificity, potentially increasing productivity especially due to their superior speed. Currently, there are several other commercially available systems that use AI, predominantly aimed at cervical cancer screening. AI assisted systems are faster than manual screening and are equivalent to manual methods in terms of lesional-cell detection rates [103].

There are also AI models aimed at non-gynecological cancer detection tasks, including thyroid, urinary bladder, lungs, breast, pleural effusion, and others [75].

### 4.3. Other Applications

The application of deep learning methods in pathology goes beyond image analysis for histopathological diagnosis. Integrating image analysis data with data derived from other sources, specifically molecular studies, further expands possibilities.

Using deep learning, histology may also be used to predict mutations and prognosis, and may even provide prospective guidance. Coudray et al. showed that CNNs can be trained to predict commonly mutated genes in lung cancer. Their study focused on adenocarcinoma and squamous cell carcinoma. The algorithm was able to predict six mutated genes (STK11, EGFR, FAT1, SETBP1, KRAS, and TP53) from histopathology images alone [104]. Furthermore, deep learning methods proved successful in predicting survival by extracting information solely from histology [105,106,107]. Deep learning algorithms were successfully applied to predict treatment response in various malignancies solely based on hematoxylin and eosin-stained slides, including but not limited to breast cancer, melanoma, and high-grade ovarian carcinoma [108,109,110,111].

## 5. Ethical and Regulatory Considerations

As image analysis becomes increasingly integrated into pathology and cytopathology, the ethical implications of these technologies cannot be neglected. The use of digital tools, machine learning algorithms, and AI-driven diagnostics in medical imaging presents challenges related to data privacy, algorithmic bias, accountability, and the potential impact on the doctor–patient relationship. The three main principles that should guide ethical AI are transparency, accountability, and governance [112].

The number of approved devices in Europe and the USA (CE-marked or FDA-approved) showed a steady and sustained increase in the past 10 years, mostly being approved in radiology, cardiology, and neurology. These specialties dominate the clinical trials scenery. Although fewer in number, pathology-related solutions are receiving increasing approval. These include, for instance, the determination of estrogen/progesterone expression in breast cancer, tumor invasion assessment, hot spot determination for various cancers, prostate cancer detection, acinar carcinoma detection in prostate cancer, and lymph node metastasis detection. All are labeled as class IVD (In Vitro Diagnostics) devices/solutions [113,114].

### 5.1. Patient Confidentiality and Data Privacy

One of the most critical ethical issues in image analysis is maintaining patient confidentiality. Digital pathology involves the storage and transmission of large volumes of patient data, including sensitive medical images. The use of cloud-based platforms and remote access systems, while convenient, increases the risk of data breaches and unauthorized access. Ensuring that patient data is anonymized before it is used for analysis and research is essential to protecting patient privacy. Data encryption and cybersecurity measures must be implemented to prevent unauthorized access to patient data [115,116,117].

The development and implementation of secure data transmission protocols, encryption techniques, and access controls are crucial to maintaining the privacy and integrity of medical imaging data throughout its lifecycle. Addressing these privacy and security challenges is paramount, as the improper handling or disclosure of patient data can have serious ethical, legal, and reputational consequences for healthcare providers and institutions.

Obtaining informed consent from patients for the use of their images in research and algorithm development is another key ethical concern. Patients should be fully informed about how their data will be used, including any potential for commercial use by companies developing image analysis software. Informed consent processes must be transparent, ensuring that patients understand the implications of data sharing and have the option to opt out if they do not wish their data to be used beyond clinical care [118,119].

### 5.2. Bias in Training Data

ML algorithms used in image analysis are trained on large datasets of histological and cytological images. However, if these training datasets are not representative of the diverse patient populations seen in clinical practice, the resulting algorithms may perform poorly on certain demographic groups, leading to biased outcomes. For instance, algorithms trained predominantly on images from one ethnic group may not generalize well to images from other ethnic groups, potentially leading to misdiagnosis or suboptimal care [120,121,122,123].

The potential for algorithmic bias raises concerns about exacerbating existing health disparities. If image analysis tools are not equitably developed and validated across diverse populations, marginalized groups may be disproportionately affected by inaccuracies in diagnosis and treatment recommendations. This issue underscores the importance of using diverse and inclusive datasets in the development and validation of image analysis algorithms to ensure they are fair and effective for all patients [124].

Another issue that needs attention is the limited generalizability of certain algorithms; in other words, they perform well with data that come from the same source as the training data but perform poorly when given ‘outside laboratory’ data [125,126]. This may hamper the wide-scale implementation of such algorithms. The normalization of images could be a solution [127,128,129,130].

### 5.3. Regulation of AI-Based Tools

Many AI-driven image analysis tools, particularly those based on deep learning, operate as “black boxes”, meaning their decision-making processes are not easily interpretable by humans. This lack of transparency raises ethical concerns about accountability, especially when these tools are used in clinical decision making. If an AI system makes an error that leads to patient harm, it may be difficult to determine how and why the error occurred, complicating efforts to assign responsibility [131]. Therefore, the practical deployment of AI-based medical image analysis tools must function within strict regulatory frameworks, such as those established by the Food and Drug Administration or the European Medicines Agency, to ensure their safety and efficacy [132,133,134].

As stated by the ASC Digital Cytology Task Force, the implementation of these technologies requires rigorous validation. More experience has been gained in surgical pathology in this regard, but as incorporation into cytopathology solutions grows rapidly, regulatory frameworks will eventually follow [103,135].

### 5.4. Human vs. Machine Debate

The integration of AI and automated tools into medical practice can negatively impact the doctor–patient relationship. Patients may be concerned about the use of such tools in their diagnosis and treatment, particularly if they feel that decisions are being made by machines rather than human healthcare providers. This concern can erode trust in the healthcare system. Proper and open communication with patients about the role of AI in their care, to emphasize that these tools support, not replace, human expertise, may lighten these concerns [136,137].

While automation does improve efficiency and reduce the burden on healthcare providers, it is crucial to maintain human oversight in the diagnostic process. Automated image analysis tools should be viewed as aids that enhance, rather than replace, the judgment of trained pathologists and clinicians. Ensuring that clinicians remain actively involved in the interpretation of AI-generated results is essential for maintaining the quality of care and addressing any errors or uncertainties that may arise [138].

Deep learning technologies have the potential to ‘zoom in’ to details of an image by reconstructing missing data, potentially adding additional data. This computational zooming should be judiciously applied to microscopic images since added data would produce undesirable outcomes [139,140].

Finally, the universal adoption of AI tools in pathology, sooner or later, will undoubtedly improve many aspects of this field of medicine, including but not limited to staff shortages. But, as one group warns, it may well cause ‘deskilling, dethrilling, and burnout’ in the practicing pathologist. These factors need to be acknowledged and addressed [141].

## 6. Interoperability and Integration Challenges

The widespread adoption of medical image analysis techniques, particularly those involving digital pathology and AI-based tools, has highlighted the importance of ensuring interoperability and effective integration with existing clinical workflows and information systems. Seamless integration is crucial for the successful translation of these advanced technologies into routine clinical practice.

One key challenge is the need to address the interoperability of image analysis platforms, image data formats, and communication protocols. Differences in software systems, data storage formats, and communication standards can hinder the exchange and sharing of medical images and the associated analysis results across healthcare institutions and clinical settings. Addressing these interoperability issues through the development of standardized data formats, open interfaces, and robust integration frameworks is essential for enabling the widespread adoption and effective utilization of medical image analysis tools.

Additionally, the integration of image analysis outputs with electronic health records, decision support systems, and other clinical information systems is crucial for ensuring that the insights derived from image analysis are readily accessible and can be effectively incorporated into the decision-making process. Overcoming the technical and organizational barriers to such integration remains an important area of focus for the field of medical image analysis.

While advancements in computational techniques have been promising, several challenges remain in the field of medical image analysis. The complex and heterogeneous nature of histopathological and cytopathological samples, as well as the need for large annotated datasets, can pose significant hurdles for the development and deployment of robust machine learning models. Unfortunately, there still remains a variability in tissue quality, staining, image quality, and scanning properties among various laboratories and systems. Nevertheless, the continuous progress in machine learning, coupled with the increasing availability of digital pathology data, offers exciting opportunities for the future of medical image analysis. Advancements in this field have the potential to enhance diagnostic accuracy, improve prognostic predictions, and support personalized treatment strategies, ultimately leading to better patient outcomes.

## 7. Future Perspectives

The field of image analysis in pathology is on the cusp of significant advancements, driven by rapid developments in AI, ML, and emerging imaging technologies. Coupled with already advanced image-processing power and large data-storage capacities, these innovations promise to revolutionize diagnostic practices, enhance the accuracy and speed of disease detection, and broaden the accessibility of healthcare globally.

### 7.1. Advancements in AI and Machine Learning

The integration of AI and ML into pathology image analysis has the potential to transform the field, enabling more precise and reliable diagnoses. With the growing availability of digital pathology data, these technologies can uncover patterns that are often imperceptible to the human eye, improving diagnostic accuracy and patient outcomes. However, the adoption of AI in clinical settings is contingent on developing models that are not only powerful but also interpretable by clinicians.

Further developments include multiple-instance learning that enables not only disease diagnosis from WSI but also the prediction of molecular alterations. To attain enhanced precision and reduced error rates, larger annotated datasets are essential [88,90].

As AI technologies continue to evolve in the future, their integration into pathology will continue and expand to broader applications. AI’s unprecedented speed and growing precision will likely lead to a more widespread use, with the technology possibly being integrated into digital slide scanners or available as cloud-based solutions. As good-quality and controlled WSI datasets are constantly being created, diagnostic accuracy and efficiency will likely increase. The availability of the Internet in low-resource regions will enable telepathology workflows coupled with AI capabilities that will allow experts from large distances to provide diagnosis and enhance overall medical care in these areas without the necessity of being there physically. One of the most exciting domains is the extraction of various data from digital slides that offer predictive information related to molecular data, treatment response, and recurrence risk. Published data indicate that deep learning methods are effective in early diagnosis and, therefore, could be used in large-scale screening such as cervical cancer and lung cancer [34,72]. Equally exciting is the integration with genomic data alongside morphological features. This could provide more comprehensive insights into diseases.

It is difficult to predict when the integration of AI tools into everyday work will become the norm, just as computers entered into people’s lives a couple of decades ago. But when they do, they will certainly reduce administrative and repetitive tasks, enabling pathologists to focus on more complex decision making and the interpretation of results [141]. Sticking to the analogy, just as computers did not replace humans, it is unlikely that some magical machine-backed system will replace pathologists. It is more probable that AI will complement pathologists, providing data-driven insights that augment human judgment. This collaborative approach will improve diagnostic confidence and patient outcomes.

### 7.2. Explainable AI

Explainable AI (XAI) is becoming increasingly critical in pathology, where the interpretability of AI models can significantly impact clinical decision making. In histology and cytology, where the visual assessment of tissues and cells is crucial, XAI tools help clinicians understand the rationale behind AI-driven diagnoses, fostering trust and adoption of these technologies in medical practice. XAI techniques, such as saliency maps and local interpretable model-agnostic explanations (LIME), highlight the features within images that influence the AI’s predictions, making the decision-making process transparent and easier to validate [142,143]. This transparency is essential in regulatory contexts, where the clinical use of AI must meet stringent standards for reliability and safety [144,145,146,147].

### 7.3. AI for Predictive and Preventive Diagnostics

Predicting cancer before symptoms appear represents, and continues to do so, a major challenge for medicine. The main purpose of identifying precursor lesions of invasive cancer is to reduce mortality and treatment-associated complications and comorbidities. The earlier the lesion is detected, the more successful is a radical and definitive treatment. In this regard, cytopathology is the flagship of early detection, boosted by its ability to obtain morphological data with minimally invasive techniques. Despite the fast development of DL models in histopathology to predict disease progression and outcome, cytopathology remains a key factor in the early detection of disease, and continues to be crucial for early disease detection due to its minimally invasive characteristics. It is too early to tell if a combination with AI capabilities would further enhance the current state-of-the-art screening potential of cytology [148,149,150,151,152].

AI’s role in predictive and preventive diagnostics is rapidly expanding, offering new possibilities for early disease detection and intervention. By analyzing large datasets, including histopathological images, AI models can identify early biomarkers of diseases such as cancer, even before they manifest clinically. This capability would allow for timely interventions that can prevent disease progression, potentially saving lives and reducing healthcare costs. Furthermore, AI-driven models can integrate diverse data types, including genetic, environmental, and lifestyle factors, to provide personalized risk assessments and preventive strategies tailored to individual patients. These advancements could shift the focus of healthcare from reactive treatment to proactive prevention [138,153].

### 7.4. Emerging Technologies

In addition to AI and ML, several technologies are emerging to enhance image analysis in pathology, pushing the boundaries of what is currently possible. These technologies include quantum imaging and super-resolution techniques, which offer unprecedented levels of detail and accuracy, and the integration of omics data with image analysis, which enables a more comprehensive understanding of disease processes.

Quantum imaging and super-resolution techniques represent the next frontier in pathology image analysis. Quantum imaging leverages quantum properties of light to achieve a higher resolution and sensitivity than conventional imaging techniques, potentially allowing for the detection of molecular-level changes in tissues [154,155,156]. Super-resolution microscopy, on the other hand, bypasses the diffraction limit of light, enabling the visualization of structures at the nanometer scale [157,158]. These technologies could provide pathologists with previously unattainable levels of detail, facilitating the identification of the subtle pathological changes that are crucial for early diagnosis and treatment.

### 7.5. The Role of Image Analysis in Telemedicine

As telemedicine continues to grow, image analysis is becoming an integral component of remote diagnostics, enabling the provision of high-quality healthcare regardless of geographical location. The ability to analyze images remotely allows pathologists to collaborate across borders, providing expertise to regions with limited access to specialist care. Remote diagnostics, facilitated by advanced image analysis tools, is transforming healthcare delivery by making expert diagnostic services accessible to patients in remote or underserved areas. Digital pathology platforms enable pathologists to analyze images from across the globe, offering rapid diagnostic services without the need for physical slides to be air-mailed. This capability not only reduces the time to diagnosis but also democratizes access to high-quality healthcare. As telemedicine expands, the integration of AI-powered image analysis into remote diagnostics will be essential in ensuring accurate and efficient care delivery worldwide [159,160].

## 8. Conclusions

There is a universal shortage of pathologists, yet workload is continuously increasing with pressure mounting to provide more precise diagnosis, often with therapeutic consequences. The integration of computer-assisted image analysis in pathology and cytopathology has transformed the field, enabling more objective and quantitative assessments of tissue and cellular features. These automated techniques can simplify the storage and sharing of digitized samples, as well as facilitate the analysis of large cohorts that would be challenging for manual review. The increasing digitization of histopathological and cytopathological samples has enabled the development of specialized software and hardware solutions for image analysis. These include digital slide scanners, image management systems, and analytical software packages that incorporate various computational techniques. These developments carry the potential to improve diagnostic accuracy, enhance prognostic stratification, and support personalized treatment planning, ultimately leading to better patient outcomes. On the other hand, they pose new challenges in terms of clinical validity, interoperability, and ethical concerns. The future of pathology and cytopathology is predicted to be marked by advancements in computer-aided image analysis. The future of image analysis is promising, and the increasing availability of digital pathology data will invariably lead to enhanced diagnostic accuracy and improved prognostic predictions that shape personalized treatment strategies, ultimately leading to better patient outcomes. As these technologies continue to evolve, ongoing ethical scrutiny and the development of robust ethical frameworks will be essential to ensure that they are used in a manner that benefits all patients.

## Figures and Tables

**Figure 1 jimaging-10-00252-f001:**
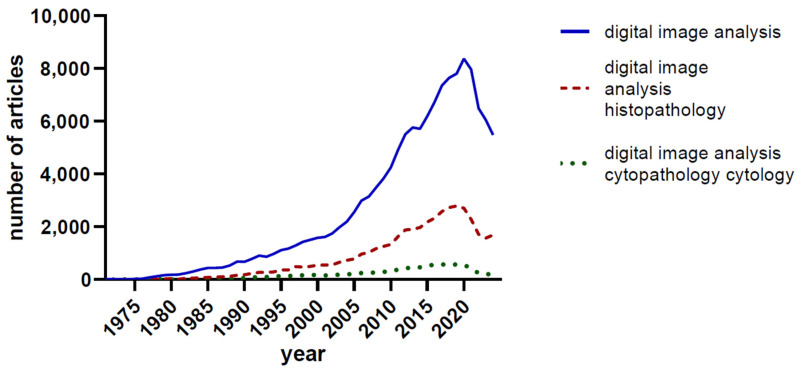
The number of PubMed-indexed articles using the search terms ‘digital image analysis’, ‘digital image analysis histopathology’, and ‘digital image analysis cytopathology/cytology’.

**Figure 2 jimaging-10-00252-f002:**
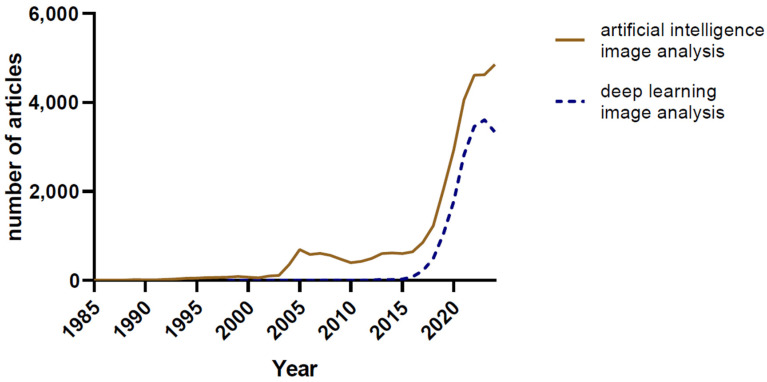
The number of PubMed-indexed articles using the search terms ‘artificial intelligence image analysis’, and ‘deep learning image analysis’.

**Figure 3 jimaging-10-00252-f003:**
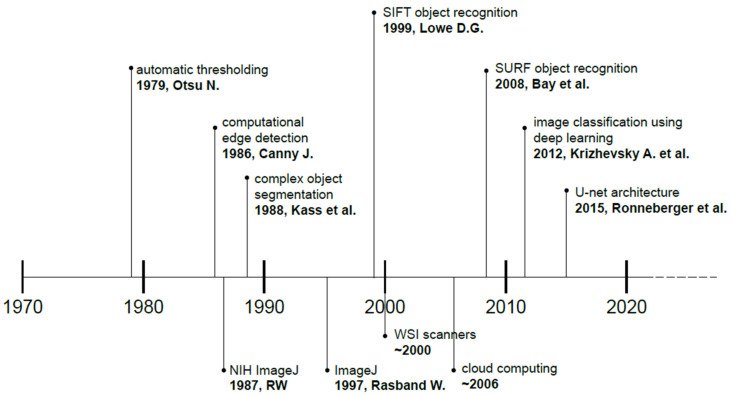
Major landmarks and publications that had a significant impact on digital image analysis (SIFT: Scale Invariant Feature Transform; SURF: Speeded-Up Robust Features; WSI: Whole Slide Image). More details and references in the text [21,22,23,24,25,26,27].

**Figure 4 jimaging-10-00252-f004:**
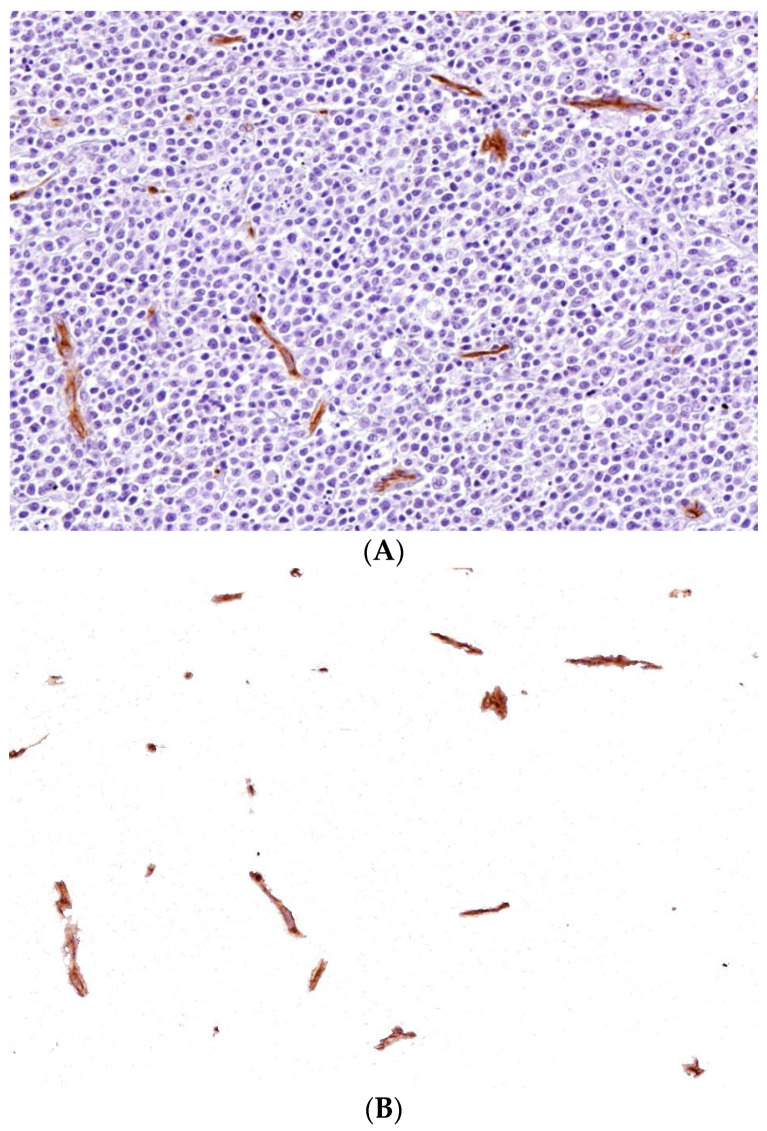
Color-based segmentation of malignant lymphoma CD34+ endothelial cells (DAB chromogen, ImageJ (v1.8.0); (**A**): Original image, before segmentation; (**B**): The result after color-based segmentation; images used with permission [51]).

**Figure 5 jimaging-10-00252-f005:**
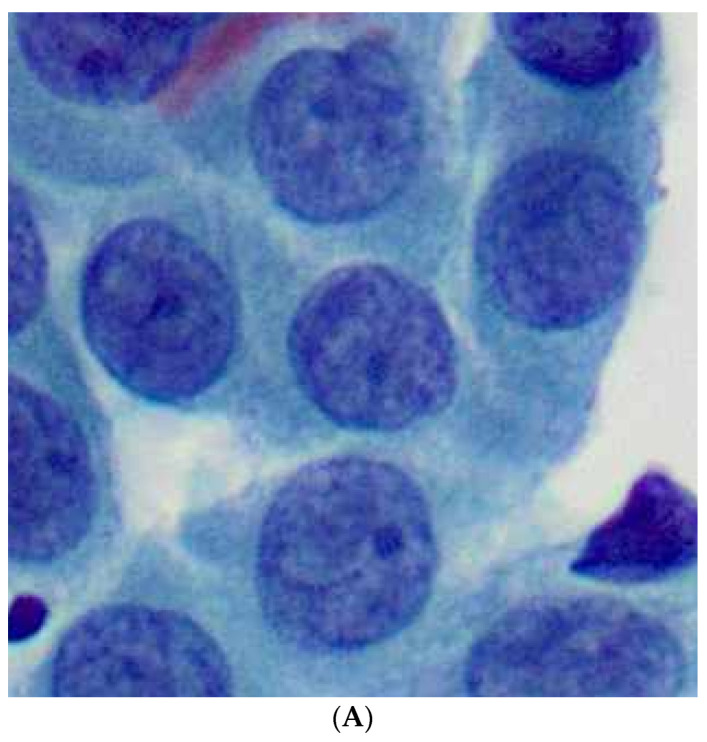
Nuclear contour tracing and thresholding using ImageJ, papillary thyroid carcinoma FNA smear, and Papanicolaou stain. (**A**): Original image; (**B**): Manual tracing of four tumor cell nuclei (1–4); (**C**): Binary image appearance after thresholding.

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
