# Peer review of "Image Analysis in Histopathology and Cytopathology: From Early Days to Current Perspectives"

_2313-433X, 2024, doi:10.3390/jimaging10100252_

Round 1
Reviewer 1 Report
Comments and Suggestions for Authors
Tibor M., et al., in their manuscript “Image analysis in pathology and cytopathology: from early 2 days to current perspectives” reported Tibor M., et al., in their manuscript “Image analysis in pathology and cytopathology: from early days to current perspectives” reported the historical evolution and current advancements in image analysis within the fields of pathology and cytopathology. They highlight how traditional manual methods, dependent on the expertise of pathologists, have transitioned towards more objective and efficient computer-aided techniques. The authors emphasize the transformative impact of digital pathology and machine learning on diagnostic processes, noting that artificial intelligence (AI) is revolutionizing the field by enabling predictive models that support diagnostic decision-making. Furthermore, they project a future where advancements in computer-aided image analysis, fueled by the growing availability of digital pathology data, will lead to enhanced diagnostic accuracy and the development of personalized treatment strategies, ultimately improving patient outcomes. However, this manuscript still has some issues that need to be resolved. The following problems should be fully addressed before further consideration:
1. In the introduction, the authors state: "Pathology is one of the more traditional branches of modern medicine, encompassing histopathology, cytopathology, and molecular pathology." However, the manuscript predominantly focuses on histopathology and cytopathology. Given that "pathology" is a broader term that encompasses "cytopathology," it is unclear why the title specifically highlights "pathology and cytopathology." The authors should either modify the title to more accurately reflect the manuscript's content or provide a justification for their choice.
2. In Section 2, paragraph 4, the authors mention: "The last decade has witnessed a surge in the application of more advanced machine learning and deep learning algorithms to the analysis of medical images, particularly in the fields of histopathology and cytopathology." It would greatly enhance the manuscript to include a visual representation, such as a chart or graph, that illustrates the progress in image analysis techniques over the past ten years. This could include data such as the increase in the number of publications and the emergence of landmark studies in the field.
3. The content in Section 3.2.3, particularly the second paragraph on quantitative features, would be more appropriately placed in Section 3.2.2. Additionally, the term "advanced computing techniques" is overly general. The authors should provide specific examples to clarify and expand on this point, thereby making the discussion more concrete and informative.
4. The sections "3.3.1: Applications of Deep Learning" and "3.3.2: AI-Driven Diagnostic Tools" appear to have considerable overlap in content. Since deep learning is a subset of artificial intelligence and the ultimate goal of pathology is diagnosis, the distinction between these two sections is unclear. The authors need to justify why these sections are separated or consider merging them to create a more cohesive narrative.
5. Section 4.1 lacks empirical examples demonstrating the superiority of models in diagnostics. This issue is prevalent throughout the manuscript. The authors should include specific data or case studies that illustrate the advantages of the discussed models in practical diagnostic scenarios, thereby strengthening their arguments.
6. While the manuscript discusses the application of deep learning in pathology diagnosis, it omits discussion on more advanced applications, such as using histology to predict mutations, survival rates, and treatment responses. Including these aspects would add depth and richness to the manuscript, offering a broader view of the potential of deep learning in pathology.
7. The discussion in Section 5 on ethical and regulatory considerations is somewhat generic and does not offer novel insights. The authors should expand on the current clinical approval stages of deep learning systems in pathology, providing a more up-to-date and relevant discussion on the practical challenges and milestones in bringing these technologies to clinical practice.
8. The manuscript predominantly discusses traditional histological staining methods. In recent years, virtual staining, a deep learning-based technique that digitally simulates histological stains, has emerged. Discussing this technique could enrich the manuscript and provide readers with insights into cutting-edge innovations in the field.
9. The last sentence of Section 7.1, "Excellent reviews are available by Song and Laak et al.," is uninformative as it merely references the reviews without discussing their content. This reduces the utility of the citation for readers. A similar issue is present in Section 3.1.1. The authors should either provide a brief overview of these reviews' content or remove the citation to avoid redundancy.
10. Section 7.1, which discusses the potential future innovations that AI could bring to pathology, should be a highlight of the manuscript. However, it is written in a rather conventional and brief manner. The authors are encouraged to expand this section, providing a more visionary and detailed discussion of how AI could transform the field in the coming years.
11. In Section 7.3, the authors mention that AI models can predict cancer diagnosis before clinical symptoms appear through the analysis of datasets, including histopathology. The authors should clarify whether such models currently exist. Additionally, considering the invasive nature of histopathological sampling, the appropriateness of using this approach for early prediction is questionable. Cytopathology, which is less invasive and often causes minimal or no harm to patients, might be a more suitable approach for early diagnosis. The authors should address this and potentially propose cytopathology as an alternative for early prediction.
Reviewer 2 Report
Comments and Suggestions for Authors
The authors discuss the pathology and cytopathology in their review, but I have major concerns:
1. A timeline illustrating the development of these techniques is needed for clarity.
2. Photos from prior research must be cited within the figures and copyright confirmed.
3. There are missing references, particularly in the introduction (lines 47-63) and headlines 1.1, historical development (lines 85-92), and headline 6. Authors should verify their sources.
4. Abbreviations should be defined upon first use; AI is repeated without this.
Reviewer 3 Report
Comments and Suggestions for Authors
The paper provides a comprehensive overview of the evolution of image analysis in the field of pathology, emphasizing both its historical context and future potential. One of the strengths of the paper is its ability to trace the progression from manual methods to modern digital tools and artificial intelligence (AI). The historical overview sets the stage for the current technological advancements, which the paper covers in depth, particularly the use of Whole Slide Imaging (WSI), machine learning (ML), deep learning, and AI-driven applications. These sections are particularly relevant in today’s medical landscape, as they offer a glimpse into how technology is reshaping diagnostics. Additionally, the paper’s discussion on ethical considerations, such as privacy concerns and algorithmic bias, is a thoughtful inclusion that demonstrates awareness of the broader implications of integrating AI into medical diagnostics.
Despite its strengths, the paper does have some weaknesses. Notably, while it touches on a wide range of topics, the discussions on specific image processing techniques such as segmentation and feature extraction remain somewhat superficial. The paper would benefit from a more detailed exploration of the methodologies used in modern image analysis, including examples of specific algorithms and their applications. Furthermore, the paper’s discussion on data challenges is somewhat lacking. The authors mention the need for large datasets but do not dive into the significant hurdles of data acquisition, such as variability in image quality, differences in staining techniques, and limited access to diverse datasets. These are critical issues in computational pathology that warrant more attention.
Another area where the paper could improve is in its balanced portrayal of AI. While the authors highlight the transformative potential of AI in diagnostic accuracy, they do not adequately address the practical challenges of integrating these technologies into clinical practice. For instance, there is little discussion on the generalizability of AI models, the infrastructure needed for widespread adoption, or the training requirements for clinicians. Additionally, the paper lacks empirical data to support some of its claims. For example, the discussion of AI’s superiority in diagnostic tasks could be increased by citing studies that provide specific performance metrics, such as sensitivity, specificity, or accuracy from real-world applications.
To strengthen the paper, several improvements should be made. First, expanding the technical sections on image segmentation, feature extraction, and classification algorithms would provide a deeper understanding for readers who are interested in the methodologies behind these technologies. Including case studies and empirical data from clinical trials or retrospective studies would also help substantiate claims about the effectiveness of AI in diagnostic settings. Furthermore, the paper should address practical challenges, such as the regulatory, infrastructural, and training barriers that may impede the integration of AI into clinical workflows. Finally, a more thorough discussion of data diversity and preprocessing techniques, such as image normalization and stain standardization, would help address one of the biggest challenges in AI-based image analysis: ensuring that models are trained on diverse, high-quality datasets.
Comments on the Quality of English Language
English is fine.
Reviewer 4 Report
Comments and Suggestions for Authors
This work is a review paper discussing the application of image analysis methods in the evaluation of microscopic images of histological preparations. Such a review is needed because there are not many similar works in this field. However, the article requires many additions before it is suitable for publication.
The contribution of this work to the field should be more precisely stated by referring to similar publications, e.g.:
10.1016/j.jasc.2023.11.006
10.1016/j.jasc.2023.11.005
10.3390/cancers14143529
10.1016/j.jasc.2024.04.003
10.1016/j.kint.2020.02.027
10.1016/j.jasc.2019.03.003
More specific examples regarding historical methods of histological image analysis should be provided, e.g.: Janowski et al., Computer analysis of normal and basal cell carcinoma mast cells. Medical Science Monitor, (2) 2001
In section 3.2. “Digital and automated image analysis”, one should provide specific algorithms, including deep learning methods used in various image processing and analysis tasks, along with an assessment of their effectiveness.
Image registration methods, which are often used as preprocessing methods for histological images, should also be included.
The number of examples in section 4 should be increased. Each example should be accompanied by a table comparing the effectiveness of different algorithms described in the literature, used for image analysis in a specific case. Then, the effectiveness and efficiency of these algorithms should be discussed.
Section 5 should consider ethical issues related to the use of patient images for research, including training deep networks. The use of such images, even after anonymization, requires the approval of the appropriate ethics committee. The informed patient consent to use their images for research is also required.
The discussion should further discuss the importance of deep learning analysis methods as the most popular approach in the analysis of a wide class of medical images.
Round 2
Reviewer 3 Report
Comments and Suggestions for Authors
The authors took into account and incorporated recommendations from reviews in this revised version.
Reviewer 4 Report
Comments and Suggestions for Authors
Thank you for proper addressing of all issues raised in my review. The paper now is suitable for publication.